# Overview of Beneficial Effects of (Poly)phenol Metabolites in the Context of Neurodegenerative Diseases on Model Organisms

**DOI:** 10.3390/nu13092940

**Published:** 2021-08-25

**Authors:** Diogo Carregosa, Sara Mota, Sofia Ferreira, Beatriz Alves-Dias, Natasa Loncarevic-Vasiljkovic, Carolina Lage Crespo, Regina Menezes, Rita Teodoro, Cláudia Nunes dos Santos

**Affiliations:** 1CEDOC, Chronic Diseases Research Centre, NOVA Medical School, Universidade NOVA de Lisboa, Campo dos Mártires da Pátria, 1169-056 Lisboa, Portugal; diogo.carregosa@nms.unl.pt (D.C.); sara.mota@nms.unl.pt (S.M.); sofia.ferreira@nms.unl.pt (S.F.); bal.dias@campus.fct.unl.pt (B.A.-D.); natasa.loncarevic@nms.unl.pt (N.L.-V.); carolina.crespo@nms.unl.pt (C.L.C.); regina.menezes@nms.unl.pt (R.M.); rita.teodoro@nms.unl.pt (R.T.); 2iBET, Institute of Experimental and Technological Biology, Apartado 12, 2781-901 Oeiras, Portugal; 3CBIOS, University Lusófona’s Research Center for Biosciences & Health Technologies, Campo Grande 376, 1749-024 Lisboa, Portugal; 4Department of Neurobiology, Institute for Biological Research “Siniša Stanković”—National Institute of Republic of Serbia, University of Belgrade, Bulevar Despota Stefana 142, 11060 Belgrade, Serbia

**Keywords:** phytochemicals, microbiota, *Saccharomyces*
*cerevisiae*, *Caenorhabditis elegans*, *Drosophila*, zebrafish, rodents, human, neurodegeneration

## Abstract

The rise of neurodegenerative diseases in an aging population is an increasing problem of health, social and economic consequences. Epidemiological and intervention studies have demonstrated that diets rich in (poly)phenols can have potent health benefits on cognitive decline and neurodegenerative diseases. Meanwhile, the role of gut microbiota is ever more evident in modulating the catabolism of (poly)phenols to dozens of low molecular weight (poly)phenol metabolites that have been identified in plasma and urine. These metabolites can reach circulation in higher concentrations than parent (poly)phenols and persist for longer periods of time. However, studies addressing their potential brain effects are still lacking. In this review, we will discuss different model organisms that have been used to study how low molecular weight (poly)phenol metabolites affect neuronal related mechanisms gathering critical insight on their potential to tackle the major hallmarks of neurodegeneration.

## 1. Neuroprotective Potential of (Poly)phenol Rich Diets

(Poly)phenols are a class of phytochemicals found on dietary sources including fruits, vegetables, and beverages like tea, wine, coffee, and cocoa [1,2]. Research on clinicaltrials.gov about the term “polyphenol” retrieves 507 clinical studies across the globe with a vast array of applications and delivery methods. These range from dietary supplementation with fruits, vegetables, seeds, or milk, for example, to the use of capsulated (poly)phenols like quercetin, resveratrol, or curcumin, showing both nutritional and pharmacological use of these molecules [3]. Epidemiological studies have demonstrated that diets rich in (poly)phenols, like the Mediterranean diet, can have beneficial effects in the brain preventing degenerative disorders and cognitive decline [4,5,6,7], modulating different aspects of synaptic plasticity like memory and/or learning in both animals and humans [5,8,9]. Long—term supplementation of (poly)phenols in animal models suggests the activation of neuronal receptors, changes in signaling pathways, and control the expression of specific genes [10]. Moreover, the use of the Mediterranean diet, with high (poly)phenolic content, is associated with a reduction of markers of inflammation, which may impact the effects on the neuroinflammation status of the brain [11].

In a large prospective cohort of older women, the consumption of berries has been related to a reduced gradual progression of cognitive decline [12]. Diet supplementation with wild blueberry was proven to improve cognitive function in older adults [13]. Moreover, chronic blueberry supplementation improved brain perfusion, task—related brain activation, and cognitive function in healthy older adults, suggesting that supplementation with an anthocyanin—rich concentrate can improve brain activation in areas associated with cognitive function [14]. In a double-blind, placebo—controlled trial, the addition of easily achievable quantities of blueberry to the diets of older adults improved some aspects of cognition [15]. Improvement on cognitive tasks was also observed in young adults using flavonol—rich cocoa while working memory improved upon using dark chocolate enriched in cocoa flavanols [16,17,18]. Nonetheless, due to the logistics, cost and difficulty to recruit human participants, human trials using (poly)phenols are mainly small in size, conducted for short periods of time or purely observational. Moreover, heterogeneity in methodologies was already identified as problematic with current evidence not supporting reliable conclusions relating to the efficacy of specific doses, duration of treatment, or sensitivity in specific populations or certain cognitive domains [19]. An overview of four meta—analyses and thirteen systematic reviews showed evidence that (poly)phenols can benefit cognition in humans, but this evidence is not as strong as it initially appears with future research requiring a more thorough analysis on the wider impact and interaction between a variety of direct and indirect mechanisms of action [19]. Meanwhile, evidences found in model organisms could help us understand such mechanisms of action behind (poly)phenol effects. Overall, the results shown in these studies support the idea that dietary habits with high content in (poly)phenols may have a positive impact on brain health reducing the susceptibility to neurodegeneration and neurodegenerative diseases, yet (poly)phenol mechanisms on neurodegeneration are still not fully elucidated.

## 2. Bioavailability of Plant–Derived (Poly)phenols and the Low Molecular Weight (Poly)phenol Metabolites

Due to their chemical similarities (poly)phenols have been classified into stilbenes, lignans, ellagitannins, flavonoids (i.e., anthocyanidins, flavonols, flavanols, isoflavones, flavones), phenolic acids (i.e., benzoic acids, cinnamic acids, propionic acids), and many others. Flavonoids are amongst the most dominant (poly)phenols present in fruits and vegetables and represent the large majority of (poly)phenols in the gastrointestinal system. Flavonoids can be found alone as aglycones, conjugated with sugars, or in the form of polymers [20]. Polymers, however, like ellagitannins, are very complex molecules with low permeability [21].

Flavonoids like anthocyanins, some isoflavones like genistein and daidzein and the flavonol quercetin can be absorbed in the stomach, reviewed in [22]. Yet, for the remaining flavonoids, absorption begins upon reaching the intestinal tract. However, several studies have reported that flavonoids and conjugates have relatively low circulating concentrations and resident times [23,24,25]. Evidence in the last two decades have shown that flavonoids conjugated with sugars are deconjugated by intestinal and microbial enzymes generating the respective aglycones. Meanwhile, flavonoid aglycones undergo catabolic reaction by the gut microbiota into smaller phenolic molecules, some of which common to those found in plants and already absorbed in initial phases of digestion, like the phenolic acids. These low molecular weight (poly)phenols still undergo phase I and II metabolic reactions by gastro–intestinal and liver cells generating the low molecular weight (poly)phenol metabolites (LMWPM) (Figure 1). The LMWPM include several dozens of different phenolic molecules that reach circulation, including benzene diols and triols, benzaldehydes, benzoic acids, phenylacetic acids, phenylpropanoic acids, cinnamic acids, hippuric acids, and many others (Figure 2). Together flavonoids and LMWPM from colon bacteria catabolism, have a higher absorption rate with more than 80% in a dose being absorbed and ultimately excreted in the urine [26,27,28]. Nonetheless, LMWPM are reported to stay in circulation for a substantial period of time, up to 48 h and reaching high concentrations of up to 50 µM, in the case of hippuric acid [29,30]. Such bioavailability of LMWPM associated with flavonoid catabolism might be perpetuated by the fact that some LMWPM, like gallic acid, ferulic acid or caffeic acid, can be present in fruits and vegetables. The presence of LMWPM in dietary sources should increase the need to investigate the consequences of LMWPM on human health. Nevertheless, although showing better ADME properties than flavonoids, staying for a longer time and being present in higher concentrations, the effects of the LMWPM is clearly under-evaluated. Additionally, once in circulation, the LMWPM distribution to target tissues is also starting to be uncovered with their ability to reach the brain becoming more evident, as recently revised [31].

## 3. Models for Studying Neurodegenerative Diseases

Neurodegenerative diseases (NDDs) represent a family of heterogeneous disorders that affect millions of people worldwide. The prevalence of these disorders is increasing due, in part, to the extension of human lifespan. NDDs are characterized by a progressive and irreversible loss of function and/or death of neurons from specific brain regions, leading to severe cognitive and functional decline. The most well—known neurodegenerative diseases are Alzheimer’s disease (AD), Parkinson’s disease (PD), Lewy body dementia, Huntington’s disease (HD) and amyotrophic lateral sclerosis (ALS), sharing common cellular and molecular mechanisms of disease, including protein misfolding and aggregation, neuroinflammation, oxidative stress and neuronal degeneration (Figure 3).

There are several forms in which intrinsically misfolded proteins exist in cells: unfolded, intermediately folded, and correctly folded species. Healthy cells degrade or try to refold misfolded proteins through chaperone proteins that are involved in protein folding and trafficking as well as intermediate stabilization. The question of the exact mechanism(s) by which protein misfolding and aggregation becomes pathological is still not answered, however, what is known is that cells lose the ability to refold or degrade misfolded proteins over time and in disease. Although different proteins are involved in the pathogenesis of different NDDs, the process of protein misfolding and aggregation is remarkably similar in all mentioned diseases (Figure 3).

In addition to protein misfolding and aggregation, a common feature of all NDDs is chronic immune activation, in particular of microglial cells, the resident macrophages of the central nervous system (CNS) (Figure 3). While an acute inflammatory response is beneficial for removing dying cells or pathogens from the brain, prolonged (chronic) inflammation has been implicated as a factor underlying neurodegeneration in several NDDs, leading to the acceleration of disease progression [33,34,35]. By releasing various kinds of damaging factors such as pro–inflammatory cytokines or proinflammatory molecules (i.e., proteolytic enzymes, reactive oxygen intermediates or nitric oxide) microglia propagate and intensify the damage to the central nervous system [36,37,38,39]. Even though great progress has been made towards understanding how neuron–glia interactions become disrupted in NDDs, many aspects of inflammation–driven neurodegeneration remain unclear. It is still under debate whether glial activation is a cause or consequence of NDDs. But even if inflammation is not a primary causative process, its presence may greatly contribute to the continued loss of neurons. Thus, the strategies to modulate the activity of microglia and astrocytes could help tilt the inflammatory balance in the right direction to treat or mitigate NDDs more effectively.

Another very important aspect of all NDDs, oxidative stress, was described as one of the major mechanisms linked to the development and progression of neurodegenerative diseases, which has made it an appealing therapeutic target (Figure 3). In neuronal cells, the primary sources of oxidative stress are dysfunctional mitochondria. Under normal conditions, endogenous antioxidants, such as glutathione (GSH) and Coenzyme Q10 (CoQ10), together with several antioxidant enzymes, such as catalase (CAT) and superoxide dismutase (SOD), are responsible for detoxifying reactive oxygen species (ROS) and reactive nitrogen species (RNS) generated by cellular processes like mitochondrial respiration. However, evidence suggests that in NDDs the activity of antioxidant enzymes is decreased, and pools of GSH and CoQ10 are depleted [40,41,42]. Loss of intrinsic antioxidant defenses allows ROS and RNS to accumulate causing oxidative damage to vital cellular macromolecules such as proteins, lipid membranes and DNA, disrupting normal mechanisms of cellular signaling [43,44]. The low activity of antioxidant defense systems, paired with the high dependence of mitochondria for energy, makes the brain extremely susceptible to oxidative stress [45,46]. There is evidence that increased oxidative stress plays an important role in the pathogenesis of neurodegenerative diseases such as AD, PD, brain ischemic disease and aging [47,48]. Data support a strong link between oxidative stress and protein aggregation processes, which are noticeably involved in the development of proteinopathies, such as AD, PD, HD, ALS and prion diseases. Moreover, the relationship between oxidative stress and inflammation has been well documented [49] with evidence indicating that oxidative stress may play a pathogenic role in chronic inflammatory diseases [50]. Thus, inhibiting oxidative stress—induced neuronal injury might represent a good strategy in the treatment or prevention of aging and neurodegenerative diseases [51].

In addition to all underlying events already mentioned, NDDs are characterized by the loss or dysfunction of groups of neurons in different brain regions specific to each NDDs. Neuronal dysfunction can be caused by neuritic dystrophy, excitotoxicity, synaptic loss and synaptic pruning, impairment of long–term potentiation or disruption of the ubiquitin—proteasome system and neuronal signaling as a result of disease pathology (Figure 3). These events often precede neuronal cell death in NDDs [52]. There is no single mechanism responsible for neuronal cell death. It is rather a combination of all the distinct aspects of NDDs mentioned above. The role of the other cell types in the CNS is crucial for neuronal homeostasis and survival. The already mentioned microglia cells and astrocytes together with microvascular endothelial cells can play several vital roles, besides inflammatory roles, like maintaining neurotransmitter, ion and energy homeostasis also conducting tissue repair and neurogenesis. As such, dysfunction in any of these well-tuned systems could be harmful to neuronal viability, leading to neuronal cell death. The blood brain barrier (BBB) integrity represents a crucial step to maintain the balance of substances in the brain and out thanks to their flow to the cerebrospinal fluid. However, it is proposed that BBB integrity might diminish during aging leading to a pathological brain microenvironment. All these factors and the increasing number of research in the last decades about the mechanisms by which neurons die in NDDs have led to the conclusion that the process of neurodegeneration is multifactorial, potentially explaining why so many drugs have so far failed to show significant results [53]. The failure to develop effective therapeutic agents for NDDs may be in large part because the majority of the current treatments target single aspects of disease pathology. As such, in recent years, efforts to develop new therapeutic strategies focusing on targeting multiple factors contributing to disease progression, rather than a single one. In this sense, dietary (poly)phenols arose as promising candidates, due to their described pleiotropic effects, since they can modulate many of the features underlying neurodegenerative processes [12].

The potential benefits of LMWPM in neurodegenerative disease processes have been shown in several brain cell lines, as reviewed in [54,55]. These cell lines have proven crucial to study some mechanisms behind the effects of LMWPM. Nevertheless, to fully explore the complex environment and mechanisms underlying neurodegenerative diseases model organisms are required, thanks to their genetic tools and libraries, and mechanistic and phenotypical similarities to humans. Several examples could be highlighted, like the genetic studies in yeast that reveal the mechanisms involved in autophagy and vesicle trafficking, two processes involved in cancer and NDDs [56,57,58]. Or the sophisticated genetic tools in *Drosophila* that allow rapid generation of models for human diseases, the evaluation of functional effects of human variant alleles and testing new therapeutic drugs [59,60]. Additionally, *Danio rerio*, the zebrafish, is a highly suitable model system for investigating gene functions involved in hematopoiesis and screening for novel potential drugs [60]. In the end, murine models of human diseases are the most common, reflecting the best genetic and physiological similarities between humans and rodents [61]. For this reason and guided by the potential of LMWPM to affect cellular processes in brain cells, we reviewed the current evidence of LMWPM studies in several models that recapitulate the main cellular hallmarks behind neurodegeneration. As such we have searched Pubmed for the list of LMWPM presented in Figure 2 and their possible synonyms [32], together with the following search terms: protein misfolding and aggregation, neuroinflammation, oxidative stress or neuronal degeneration; and the organism: yeast, *Drosophila*, *C. elegans*, zebrafish or rodents (mice and rat) thereby showing the results herein.

## 4. Low Molecular Weight (Poly)phenol Metabolites in *Saccharomyces cerevisiae* Models of NDDs

The budding yeast *Saccharomyces cerevisiae* is one of the most studied organisms in the eukaryotic kingdom. Presenting unique attributes, such as easy handling, short generation time and high amenability for genetic manipulation, *S. cerevisiae* has been established as a versatile and powerful experimental model in diverse areas of biology [62,63]. Remarkably, yeast shares highly conserved molecular and cellular mechanisms with human cells (e.g., cell division, DNA replication, metabolism, protein folding and intracellular trafficking), thus representing an invaluable model to study the major biological processes involved in several human diseases [64]. More than 400 essential yeast genes can be replaced with their human orthologs. Moreover, nearly a thousand yeast genes belong to orthologous gene families associated with human diseases [65]. In what concerns NDDs, yeast models have been described to recapitulate important pathological features, with a special focus on mitochondrial dysfunction, trafficking defects, proteasomal impairment and transcriptional dysregulation [64,66]. Besides being useful for the study of NDDs intricacies, such models have also been explored as in vivo platforms to identify the bioactive properties of compounds with therapeutic potential [63]. The data presented in this section summarizes the knowledge generated using yeast as a simple eukaryotic model to screen the biological effects and respective cellular targets of LMWPM in the protection against the pathological mechanisms of NDDs.

ATP—binding cassette (ABC) transporters are described to be involved in the maintenance of brain homeostasis by extruding toxic peptides (e.g., Aβ) that accumulate in the brain during disease progression [67]. The reduced function of these efflux pumps has been noted in most NDDs [68]. In *S. cerevisiae*, LMWPM were shown to influence these transporters (Table 1). The metabolite 3,4,5–trihydroxybenzoic acid, for instance, weakly inhibited the plasma membrane Pdr5 efflux pump in AD124567 yeast strain overexpressing the *PDR5* gene [69]. In another study, benzoic acid induced the expression of Pdr12, an ABC transporter catalyzing the active efflux of weak acids under stress conditions [70]. In BY4741, increased resistance to 2—hydroxybenzoic acid was mediated by overexpression of efflux pump—encoding genes, including *YOR1*, *SNQ2*, *AZR1*, and *FLR1* [71]. Either Pdr5 and Pdr12, as well as Snq2, are protein orthologues of human ABC A—subfamily transporters which have been linked to CNS integrity maintenance, brain lipid homeostasis and neurodegenerative disorders [72].

Additional studies have revealed the effect of LMWPM towards autophagy and translation, two mechanisms described to be perturbed in NDDs (Table 1). Studies in more complex systems indicate that in NDDs, cells exhibit defects in multiple steps of RNA processing and protein synthesis [73]. Mutated SOD1 and pathological tau were described to be present in inclusions containing core components of stress granules, thus affecting their dynamics and possibly interfering with the translation process [73,74]. According to Hazan et al., benzoic acid inhibited macroautophagy, induced the maturation of prApe1 and impaired ER–to–Golgi transport in yeast [75]. Another study demonstrated that 4—hydroxy–3—methoxybenzaldehyde represses translation in yeast as concluded by the accumulation of processing bodies (P—bodies) and stress granules composed by non—translating mRNAs and proteins after 4—hydroxy–3—methoxybenzaldehyde exposure [76]. In the *zwf1* mutant for cytoplasmic glucose–6—phosphate dehydrogenase involved in adapting to oxidative stress, 4—hydroxy–3—methoxybenzaldehyde also increased the P—bodies formation [77].

Yeast models of protein aggregation are usually based on the heterologous expression of a specific human protein fused to functional domains and fluorescent tags, such as GFP [62,78]. Despite the large availability of AD, PD, ALS and HD yeast models amenable for screening purposes and mechanistic studies [62,63,64,79,80], few studies address LMWPM bioactivity. Using a yeast model of AD, Porzoor et al. screened a set of LMWPM for activity against Aβ_42_ oligomerization (Table 1). The authors used a GFP fusion system in which the percentage of fluorescence emission is dependent on the correct folding of GFP as a reporter for Aβ_42_ aggregation. Among the tested compounds, 3,4,5—trihydroxybenzoic acid as well as benzoic acid and 4—hydroxy–3—methoxycinnamic acid were shown to reduce GFP fluorescence and in vitro aggregation of GFP—Aβ_42_ fusion [81].

In the presence of H_2_O_2_, benzene–1,2,3—triol mediated the reduction of intracellular oxidation and protein carbonylation levels thereby contributing to oxidative stress resistance. Evaluation of benzene–1,3,5—triol under the same conditions did not show any protection. The effect of benzene–1,2,3—triol was independent of the modulation of endogenous antioxidant defenses, without alterations in superoxide dismutase and catalase basal activity. Despite this protection, benzene-1,2,3-triol did not restore the decreased chronological lifespan associated with exacerbated oxidative stress in a mutant lacking superoxide dismutase gene. These data indicate the inefficiency of benzene–1,2,3—triol to protect aged cells suffering oxidative damage, at least in the absence of *SOD2* [82]. In contrast, Sunthonkun et al. have shown the positive effects of 3,4—dihydroxybenzoic acid during chronological aging in yeast. In this study, 3,4—dihydroxybenzoic acid effectively modulated life span—extension by reducing ROS and by conferring the cells an improved resistance to free radicals (Table 2). According to the authors, concerning the reduction of ROS levels 3,4—dihydroxybenzoic acid seems to mimic the inactivation effect of Sir2, Tor1 or Sch9 while for resistance to free radicals it seems to compensate for *RIM15*, *MSN2*, *MSN4* or *ASG1* deletion [83].

LMWPM can also function as prooxidants inducing the increase of ROS and cellular antioxidant defenses. In this perspective, Nguyen et al. demonstrated that 4—hydroxy–3—methoxybenzaldehyde increased intracellular oxidation, and induced the expression of antioxidant machinery through the activation of Yap1, the master regulator of yeast oxidative stress responses. Moreover, 4—hydroxy–3—methoxybenzaldehyde also facilitated mitochondria fragmentation [84]. In another study (Table 2), benzene–1,2—diol appeared to disrupt intracellular vesicle trafficking as an indirect consequence of lipid peroxidation triggered by cytoskeletal disruption and/or oxidative stress. The authors did not identify the requirement of mitochondrial electron transport chain, suggesting that stimulation of H_2_O_2_ production by benzene–1,2—diol does not significantly contribute to oxidative toxicity [85]. At the millimolar concentration range, 4—hydroxybenzoic and 4—hydroxy–3—methoxycinnamic acid were also shown to increase ROS cytosolic levels and to affect the growth of WT cells (Table 2) [86]. The deletion of *BNA7* gene, encoding a Kynurenine formamidase, was reported to improve the growth of these cells in the presence of 4—hydroxy–3—methoxycinnamic acid by a mechanism not related to ROS reduction. Evidence has shown that yeast deletion mutants for the tryptophan catabolic pathway can suppress the toxicity of NDDs—related protein aggregation, such as huntingtin [87]. Interestingly, through a genomic screen in cells subjected to 4—hydroxy–3—methoxycinnamic acid, the authors identified several pathways hyper-sensitive to protein aggregation (i.e., lipid metabolism and vacuolar protein sorting), suggesting a possible link between *BNA7* deletion and the reduction of protein aggregation during exposure to this metabolite [86].

Although studies in yeast have largely contributed to the knowledge on the potential role of dietary (poly)phenols in protection against NDDs, in contrast, identification and characterization of the molecular mechanisms of LMWPM using these models are still poorly explored. Most of the existing studies using *S. cerevisiae* focus on the protective effects conferred by natural extracts or pure flavonoids [63], excluding the real bioactive effectors of such response. Additionally, the great majority of these studies evaluate the protective role of metabolites in a non—specific manner (i.e., chemical induction of oxidative stress) to address NDDs—related molecular pathways and do not explore yeast models recapitulating specific disease features.

Noteworthy, modelling NDDs in a single—cell organism has some limitations that must be considered. Yeast fails to capture organism-level phenotypes emerging from inter—cellular communication [88]. On the other hand, and specifically in the context of NDDs, it does not display neuro—typical morphological structures that are indispensable for correct cell functioning [89]. For those reasons, all evidence provided by yeast models should be carefully interpreted and further validated with studies in more complex systems. Even so, being a unicellular organism bares the complexity of higher eukaryotic cells, yeast represents a robust model to study the cellular events without the interference of complex interactions characteristic of the nervous system cells [90]. By directly studying the endogenous protein orthologues of human counterparts or promoting the heterologous expression of human disease—related proteins, yeast models constitute a valuable tool to discover and dissect the molecular pathways involved in human disorders. The simplicity of yeast models makes them a powerful first—line screening system for rapid and efficient discovery of novel leads and therapeutic targets for human diseases.

**Table 1 nutrients-13-02940-t001:** Effects of low molecular weight (poly)phenol metabolites on the indicated hallmarks of NDDs in *Saccharomyces cerevisiae*.

(Poly)phenol Metabolite ^1^	Model	Dose and Time	Mechanistic Evidences	Ref.
Benzenes				
Benzene–1,2—Diol (Catechol)	WT and Several Mutants (BY4743) for Oxidative Stress Resistance, Cytoplasmic Thioredoxin System, GSH Synthesis and Recycling, Lipid Metabolism, Vacuole and Endosomes Biogenesis, V–ATPase Assembly and Alpha–Tubulin Folding.	0.55 mM, 1.1 mM and 2.2 mM for 5 or 15 Generations	↑ Response Pathways Associated with Oxidative and Cytoskeleton Stress, ER Stress and Vesicular Trafficking	[85]
Benzaldehydes				
4—Hydroxy–3—Methoxybenzaldehyde (Vanillin)	WT (YPH250)	4, 6 and 8 mM for 30 min	↑ *ZWF1*, *GND1* and *GND2* Expression Levels	[77]
	*zwf1* Mutant (YPH250)	2–8 mM for 15 min	↑ Mitochondrial Fragmentation	
		10–20 mM for 30 min	↑ P–Body Formation	
	WT and Mutants for Chromatin Remodeling, Vesicle Transport, Cell Cycle, DNA Processing and Cellular Transport (BY4743)	5 mM for 24 h	↓ Cell Growth	[77]
	WT (BY4741)	4 mM for 6 h	↓ Translation Activity	[76]
		2–50 mM for 30 min	↑ P–Body Formation	
		30–50 mM for 30 min	↑ Stress Granules Formation	
	WT (BY4742)	6–15 mM for 30 min	↓ Translation Activity	[91]
		6–15 mM for 1 h	↑ *ADH6* and *ADH7* mRNA Levels	
	*adh6* and *adh7* Mutants (BY4742)	8 mM for 36 h	↓Growth Rate	
Benzoic Acids				
3,4,5—Trihydroxybenzoic Acid (Gallic acid)	WT and *zrt1* Mutant (BY4741)	100 µM for 1 h	↓ Intracellular Zn Content ↑ *ZRT2* Transcription	[92]
	Pdr5 Overexpressing Yeast (AD12456)	5–200 µM for 1 h	Inhibited the Pdr5 Efflux Pump	[69]
	*ahp1* Mutant (BY4743) Overexpressing Aβ_42__GFP	50 µM for 6 h	↓ Protein Aggregation	[81]
2—Hydroxybenzoic Acid (Salicylic Acid)	H208-3B Carrying PGAPAQ1 Plasmid	5 mM	↑ [Ca^2+^] Cytosolic Levels	[93]
	*yrr1* Mutant (BY4741)	12.5 mM	↑ Resistance to 2HBA Mediated by Efflux Pump–Encoding Genes Overexpression	[71]
Benzoic Acid	WT (FY1679–28c)	0.5 mM for 1 h	↑ Pdr12 and *HSP30* Expression	[70]
	*vac8* Mutant (HAY394)	0, 0.1, 0.5, and 1 mM for 2 h	↑ prApe1 Maturation	[75]
		2 mM for 2 h	↓ Macroautophagy	
	WT (HAY75)	2 mM for 5, 15 and 40 min	↓ ER–to–Golgi Complex Transport Step	
2,5—Dihydroxybenzoic Acid	*ahp1* Mutant (BY4743) Overexpressing Aβ_42__GFP	50 µM for 6 h	↓ Protein Aggregation	[81]
2,3—Dihydroxybenzoic Acid				
3,4—Dihydroxybenzoic Acid				
Phenylacetic Acids				
Phenylacetic Acid	*pdr12* Mutant (CEN.PK113–7D)	2–8 mM for 72 h	↓Cell Growth	[94]
Cinnamic Acids				
4—Hydroxy–3—Methoxycinnamic Acid (Ferulic Acid)	WT (BY4741)	1 mM for 1 h	↑ YAL005C, YER153C, and YPR125W Expression	[95]
	*ahp1* Mutant (BY4743) Overexpressing Aβ_42__GFP	50 µM for 6 h	↓ Protein Aggregation	[81]
2,4—Dihydroxycinnamic Acid	*ahp1* Mutant (BY4743) Overexpressing Aβ_42__GFP	50 µM for 6 h	↓ Protein Aggregation	[81]
3,4—Dihydroxycinnamic Acid				
3,4,5—Trimethoxycinnamic Acid				

Aβ_42_—Amyloid beta 42; *AHP1*—Alkyl HydroPeroxide reductase 1 gene (involved in cellular response against oxidative stress); GFP—Green Fluorescent Protein; *ADH6*—Alcohol Dehydrogenase 6 gene; *ADH7*—Alcohol Dehydrogenase 7 gene; ER—Endoplasmic Reticulum; *GND1*—6—phosphoGlucoNateDehydrogenase 1 gene (involved in cellular response against oxidative stress); *GND2*—6—phosphoGlucoNateDehydrogenase 2 gene; GSH—Glutathione; *HSP30*—Heat Shock Protein 30 gene; P–body—Processing body; *PDR12*—Pleiotropic Drug Resistance 12 gene; *PDR5*—Pleiotropic Drug Resistance 5 gene; prApe1—precursor amminopeptidase I; *VAC8*—VACuole related 8 gene; WT—Wilde type; *YAL005C*—Stress–Seventy subfamily A; *YER153C*—PETite colonies; *YPR125W*—Yeast LETM1 Homolog; *ZRT1*—Zinc–Regulated Transporter 1 gene; *ZRT2*—Zinc-Regulated Transporter 2 gene; *ZWF1*—ZwischenFerment 1 gene (Cytoplasmic glucose–6—phosphate dehydrogenase involved in adapting to oxidative stress).^1^ (Poly)phenol metabolites are named accordingly the recommendations recently published [32] however, the name cited in the original publications where the effect is described is indicated in brackets. ↑—*increased* ↓—*decreased*.

**Table 2 nutrients-13-02940-t002:** Effects of low molecular weight (poly)phenol metabolites on *Saccharomyces cerevisiae* cells submitted to oxidative stress or in oxidative stress—related mutants.

(Poly)Phenol Metabolite ^1^	Model	Dose and Time	Mechanistic Evidence	Ref.
Benzenes				
Benzene–1,2,3—Triol (Pyrogallol)	WT (BY4741) + 1.5 mM H_2_O_2_ for 1 h	300 μM for 15 min	↑ Cell Viability ↓ H_2_O_2_–Induced Intracellular Oxidation ↓ Protein Carbonylation ↓ Total GSH Levels	[82]
	*sod2* Mutant (BY4741) + 1.5 mM H_2_O_2_ for 1 h	300 μM for 15 min	↑Oxidative Stress Resistance	
	*sod2* mutant (BY4741)	300 μM for 4 Days	=Lifespan	
Benzene–1,3,5—Triol (Phloroglucinol)	WT (BY4741) + 1.5 mM H_2_O_2_ for 1 h	300 μM for 15 min	=Cell Viability =H_2_O_2_–Induced Intracellular Oxidation =Protein Carbonylation ↓ Total GSH Levels	[82]
	*sod2* mutant (BY4741)	300 μM for 4 days	=Lifespan	
Benzaldehydes				
4—Hydroxy–3—Methoxybenzaldehyde (Vanillin)	WT (YPH250)	4, 6 and 8 mM for 30 min	↑ Expression of Oxidative Stress Response Genes	[84]
		8 mM for 15 min	↑ Mitochondrial Fragmentation	
		8 mM for 30 min	↑ Oxidation Levels	
Benzoic Acids				
3,4—Dihydroxybenzoic Acid (Protocatechuic Acid)	WT (BY4742) + 5 mM H_2_O_2_ for 1 h	13 µM for 30 Days	↓ Oxidative Stress ↑ Yeast Lifespan	[83]
	*sir2* Mutant (BY4742) + 5 mM H_2_O_2_ for 1 h	13 µM for 17 Days	↓ ROS Accumulation	
		13 µM until OD_600_ 0.6	↑ Yeast Growth	
	*tor1, sch9, rim15, msn2, msn4 and asg1* Mutants (BY4742) + 5 mM H_2_O_2_ for 1 h	13 µM for 17 Days	↓ ROS Accumulation	
4—Hydroxybenzoic Acid	WT (BY4741) and *bna7* Mutant (YKB4649)	50 mM for 2 h	↑ Cytoplasmic ROS Levels	[86]
2,5—Dihydroxybenzoic Acid	WT and *sod2* Mutant (BY4741)	3–18 mM for 7 Days	↓ Cell Growth	[96]
	WT and *sod2* Mutant (BY4741) + 1, 2 and 3 mM H_2_O_2_ for 7 days	1 and 3 mM for 7 Days	Recovery from Oxidative Stress	
	WT, *glr* and *sod2* Mutants (BY4741) + 0.1 mM GSH for 7 Days	1 and 3 mM for 7 Days	Recovery of Cell Growth	
Cinnamic Acids				
4—Hydroxy–3—Methoxycinnamic Acid (Ferulic Acid)	WT (BY4741)	10 mM for 72 h	↓ Cell Growth	[86]
		10 mM for 2 h	↑ Cytoplasmic ROS Levels	
	*bna7* Mutant (YKB4649)	10 mM for 72 h	↑ Cell Growth	
		10 mM for 2 h	↑ Cytoplasmic ROS Levels	

*ASG1*—Activator of Stress Gene 1 (regulator involved in the stress response); *BNA7*—Kynurenine formamidase 7 gene (involved in de novo NAD biosynthesis); GSH—Glutathione; H_2_O_2_—Hydrogen peroxide; *MSN2*—Multicopy suppressor of SNF1 Stress-responsive transcriptional activator) mutation gene (; *MSN4*—Multicopy suppressor of *SNF1* mutation gene; OD_600_—Optical Density at 600 nm; *RIM15*—Regulator of *IME2* gene; ROS—Reactive Oxygen Species; *SCH9*—Serine/threonine–protein kinase SCH9 gene; *SIR2*—Silent Information Regulator 2 gene (regulator of autophagy and mitophagy); *SOD2*—Superoxide Dismutase 2 gene; *TOR1*—Target Of Rapamycin 1 gene; WT—Wilde type. ^1^ (Poly)phenol metabolites are named accordingly the recommendations recently published [32], however, the name cited in the original publications where the effect is described is indicated in brackets. ↑—*increased* ↓—*decreased*.

## 5. Low Molecular Weight (Poly)phenol Metabolites in Invertebrate Models of NDDs

Invertebrate models, such as the nematode *Caenorhabditis elegans* or the fruit fly *Drosophila melanogaster*, are two of the most used model organisms in biomedicine, where the study of the potential of small molecules to counteract NDDs is rising. These are ideal systems, because they allow the fast evaluation of behavior and the dissection of molecular mechanisms in the context of aging and in disease models.

Some of the advantages of these models are their short generation time and lifespan, the ability to yield a large progeny, conserved molecular pathways, accessible nervous system, and low maintenance costs. The aging process of these models reproduces the hallmarks of vertebrate aging, but given their much shorter lifespans, with *C. elegans* living around 21 days and *Drosophila* about 70 days at 25 °C, the study of age—associated disorders becomes much more efficient [97,98]. Therefore, the use of *C. elegans* and *Drosophila* for fast high—throughput and high—content screenings for drugs or food—derived neuroprotective compounds is growing. In addition, the large number of genetic tools available, easy genetic manipulation and less genetic redundancy allow the discovery of the molecular mechanistic routes of action of these compounds, as well as the specific cell type(s) on which they act upon. These small molecules can be easily added to the worm’s or flies’ food or directly injected into their circulation (in the case of *Drosophila*) ensuring complete absorption of the specific compounds. Their effects can be easily tracked and evaluated over time through behavioral, cellular and biochemical assays. Moreover, the immediate molecular effect of these compounds can also be observed through ex vivo acute exposure either in fixed tissues or through live imaging. In this way, both *C. elegans* and *Drosophila* can substitute for in vitro approaches, also giving the advantage of mimicking the whole—tissue and organismal responses.

### 5.1. C. elegans

Several studies used the nematode *C. elegans* to test the effect of bioactive compounds in neurodegeneration phenotypes or lifespan extension and in particular for the effects of LMWPM (Table 3). When administered to control strains of *C. elegans* 2—hydroxy–4—methoxybenzaldehyde *per se* did not induce any locomotion phenotypes. However, if administered to worms exposed to 6—hydroxydopamine (6—OHDA), a validated model of PD—like features that display slow and progressive degeneration of the dopamine system, this LMWPM can rescue some of the PD phenotypes [99]. After 6—OHDA administration, worms display a decrease in body bends compared to controls, a phenotype associated with 6—OHDA–induced PD. Furthermore, in the same study, in the transgenic strain BZ555, in which dopaminergic neurons are tagged with GFP, co—exposed to 6—OHDA and 2—hydroxy–4—methoxybenzaldehyde, it was observed that the treatment conferred neurological protection, measured as the decrease of dopaminergic cell death and rescue of the locomotion phenotype [99].

To test the effect of 3,4,5—trihydroxybenzoic acid in *C. elegans*, a wild type (ATX3Q17—GFP) and a pathological (ATX3Q130—GFP) variant of Ataxin 3, which is involved in polyQ disorders due to CAG expansions (e.g., Machado Joseph’s disease), were expressed in the nervous system under the control of the unc—119 promoter and compared with wild type Bristol N2 strain which was used as a control. After 24 h and 48 h of post—exposure, 3,4,5—trihydroxybenzoic acid was able to rescue some of the motility defects induced by the over—expression of pathogenic ATX3Q130—GFP, but no survival increase was observed [100].

Treatment of *C. elegans* with cinnamic acid has also been performed. In particular, administration of hydroxycinnamic acid [101], 3,4—dihydroxycinnamic acid [102,103], 4—hydroxy–3—methoxycinnamic acid [104] and 3–(3′,4′—dihydroxyphenyl)propanoic acid [103] have been shown to give promising results regarding the neuroprotective potential of these compounds. Cinnamic acids described as LMWPM, 3—hydroxycinnamic acid, 2—hydroxycinnamic acid, cinnamic acid, 3,4—dihydroxycinnamic acid, 2,4—dihydroxycinnamic acid, and 3,4,5—trihydroxycinnamic acid were shown to inhibit Aβ_40_ fibrillogenesis and reduced Aβ_40_—induced cytotoxicity in a dose—dependent manner in cells, demonstrating the potential of cinnamic acids and the need to test other LMWPM. When hydroxycinnamic acid was given to a *C. elegans* transgenic line, CL4176, which expresses human Aβ in muscle and gives rise to a paralysis phenotype in the worms due to the generation of Aβ deposits, the results showed that all small molecules delayed the progress of Aβ—induced paralysis in a dose—dependent manner [101].

**Table 3 nutrients-13-02940-t003:** Effects of low molecular weight (poly)phenol metabolites on models of neurodegeneration in *C. elegans.*

(Poly)Phenol Metabolite ^1^	Model	Dose and Time	Mechanistic Evidence	Ref.
Benzaldehydes				
2—Hydroxy–4—Methoxybenzaldehyde	BZ555 Worms (dat1::gfp)	2.5 mM Administered on Food for 24 h	↓ Dopaminergic Neurodegeneration	[99]
Benzoic Acids				
3,4,5—Trihydroxybenzoic Acid (Gallic Acid)	Amyloid–Related Neurodegenerative Disease Model Expression of ATX3 in the Nervous System	0.1 mM Administered to Food Scoring After 24 and 48 h	↑ Mobility No Rescue in Lifespan	[100]
Cinnamic Acids				
4—Hydroxy–3—Methoxycinnamic Acid (Ferulic Acid)	Alzheimer’s Model (Muscle–Specific Aβ—Expression)	50 mM Administered on Food from Embryo	↑ Survival Rate ↑ Protection against Aβ-Cytotoxicity	[104]
3,4—Dihydroxycinnamic Acid (Caffeic Acid)	Wt Mt(Mev-1), A Strain Hypersensitive To Oxidative Stress	100 µM to 600 µM (Strongest Effect at 300 µM)	↑ Lifespan WT and Mt(mev–1) ↓ Body Size Altered Lipid–Metabolism Tendency to Delay Reproductive Time = Number of Offspring ↓ Gene Expression of Hsp–3, Hsp–17, and Hsp–16.41 ↑ Gene Expression of Hsp–12.6	[102]
Cinnamic Acid	Alzheimer’s Model Strain CL4176 Expressing Human Aβ	40, 80, or 200 μM Throughout Development	↓ Aβ_40_ Fibrillogenesis ↓ Aβ_40_–Induced Cytotoxicity Ameliorate AD–Like Symptoms of Worm Paralysis	[101]
2—Hydroxycinnamic Acid				
3—Hydroxycinnamic Acid				
2,4—Dihydroxycinnamic				
3,4,5—Trihydroxycinnamic Acid				

WT—wild type; Aβ—Amyloid beta 42; ATX3—Ataxin 3; Hsp—heat shock protein; dat1—dopamine transporter 1. ^1^ (Poly)phenol metabolites are named accordingly the recommendations recently published [32], however the name cited in the original publications where the effect is described is indicated in brackets. ↑—*increased* ↓—*decreased*.

### 5.2. Drosophila

*Drosophila* is progressively emerging as a model organism to study the role of (poly)phenol metabolites in life extension and protection in aging and NDDs models (Table 4). Flies that were fed with 3,4,5—trihydroxybenzoic acid together with a sucrose solution, were shown to have protective properties through the increase of antioxidant enzymes activity, namely SOD, catalase and glutathione S—transferase. Furthermore, 3,4,5—trihydroxybenzoic acid has also been shown to be protective against urethane—induced genotoxicity and oxidative stress, an environmental mutagenic [105]. Additionally, 0.1 mM 3,4,5—trihydroxybenzoic acid, 0.1 mM 4—hydroxy–3—methoxycinnamic acid and 0.5 and 1 mM of 3,4—dihydroxycinnamic acid, all LMWPM, were given individually in a 0.1% sucrose solution to flies exposed to paraquat (PQ), a chemical used to induce Parkinson’s disease, and were able to extend the lifespan of the flies, and to ameliorate PQ—nduced locomotor defects [106,107]. In a different study, in which parkin (a PD—disease gene with ubiquitin—ligase activity, required for mitophagy) was knocked down in dopaminergic neurons and chronically exposed to PQ, a model in which mitophagy is impaired and ROS are produced at much higher levels, 3,4,5—trihydroxybenzoic acid at 0.1 and 0.5 mM could not rescue neither the locomotor defects nor the reduced lifespan of these flies. This result suggests that this compound does not interfere with the molecular mechanisms affected in this model, namely mitophagy and parkin [108]. Still, in a genetic model of Alzheimer’s disease 3,4,5—trihydroxybenzoic acid showed anti-BACE1 and anti—cholinesterase activity [109].

Supplementation with 3,4—dihydroxycinnamic acid (1.1 mM to 5.55 mM) significantly extended lifespan, increased climbing index, increase SOD and catalase activity, counterbalanced the aging-induced amino acid imbalance and improved mitochondrial function by regulating redox state and mitochondrial metabolism [110].

Several other studies with the dietary (poly)phenol parent compounds individually or in extracts, also suggested that these compounds are neuroprotective, likely playing antioxidant and antiaging roles. For example, the effects of green tea, apple, blueberry and black rice extracts which all contained a complex mixture of phenolic compounds have all showed an enhancement of *Drosophila* WT lifespan [111,112,113,114]. Additionally, the supplementation with green tea extract reduced mutant Huntingtin—induced neurodegeneration and positively increased the longevity in this model [115]. Furthermore, glutathione (0.16 mM), catechin (0.17 mM) and epicatechin (0.17 mM) individually fed rescued lifespan of PQ treated flies [116]. Epicatechin (0.1–8 mM) and epigallocatechin—gallate alone could increase the lifespan of WT flies [117,118]. In a transgenic model of PD epicatechin—gallate showed dose—dependent significant protection against oxidative stress and apoptosis in the brain of the flies as well as a delay in the loss of locomotor activity [119]. Likewise, propyl gallate (0.1 mM) and epigallocatechin—gallate (0.1 mM and 0.5 mM) could increase locomotor activity and lifespan of a genetic model of PD exposed to PQ [108,120].

Although informative, the use of extracts or (poly)phenol mixtures, makes it difficult to precise the potential bioactivity of each individual compound. Furthermore, the studies with dietary (poly)phenols and extracts do not take into consideration the fact that (poly)phenols might have low bioavailability in humans together with their metabolism and transformation by human microbiota into LMWPM, before reaching circulation [121].

Altogether, *Drosophila* is rising as a good model system in which to study the side effects and behavioral outputs of (poly)phenol metabolites treatment, yet the discovery of the mechanisms of action of these physiological relevant metabolites is still underappreciated. The utilization of the genetic toolkit available in this model will help the identification of the precise molecular mechanisms through which these compounds function in cells.

## 6. Effects of Low Molecular Weight (Poly)phenol Metabolites in Zebrafish Models of NDDs

Zebrafish (*Danio rerio*) is rapidly becoming a popular model for translational research on human NDDs. The neuroanatomy of the zebrafish brain resembles that of humans [122,123] and there is conservation of neurochemical and neuro-adaptation pathways [124,125,126]. Also, the architecture of zebrafish CNS shares the organization with higher vertebrates [123] while physiological, emotional and social behavioral similitudes with humans have been recognized in zebrafish [127,128,129,130]. Notably, zebrafish models of brain disorders, including Alzheimer’s Disease (AD) and Parkinson’s Disease (PD) have been successfully generated in adult and larvae [131,132,133,134]. Owing to their fairly simple nervous system and distinctive optical translucency until larval stages, zebrafish offers the unique advantage for real—time non—invasive intravital imaging of neurological processes [135] and in vivo neurophysiological and functional analysis of disease condition [136,137]. Zebrafish have large fertility and fecundity rates; therefore, are highly suitable for high—throughput and high—content screenings aiming at identifying novel neuromodulator compounds for NDDs [138,139]. Embryos and larvae are particularly amenable due to their small size, simple manipulation and observation, together with the fact that molecules can be added directly into their liquid media and rapidly uptake [140]. A good example of the use of zebrafish as a high—throughput and high—content screening platform that includes already (poly)phenol compounds was recently published for searching anti—inflammatory molecules for inflammatory bowel disease [140].

Among other model organisms, zebrafish is increasingly utilized as a powerful model for pharmacogenetics and neuropharmacology [141]. Yet, the identification of (poly)phenol metabolites using zebrafish presents a still unexplored field of research for generating new lead compounds for NDDs. Evidence of neurological modulation by the LMWPM in zebrafish neurophysiological functions (Table 5) or in zebrafish models of NDDs (Table 6) resumes to a small number of molecules. For instance, brief exposure to 3,4,5—trihydroxybenzoic acid in larval zebrafish induced motoneuronal hyperexcitability and hyperactive behavior (Table 5) [142]. The increased locomotor activity provoked by 3,4,5—trihydroxybenzoic acid was accompanied by the augmented expression of neuronal activity markers like *fosab* (ortholog for Human *c—fos* in zebrafish) in specific CNS areas, and by the acute impairment of GABAergic—glutaminergic metabolism in the larval brain (Table 5) [142]. Although the underlying mechanisms must be still clarified, these results suggest that 3,4,5—trihydroxybenzoic acid acts as an excitatory molecule, that is able to induce specific nerve responses. In the adult zebrafish brain, 3,4,5—trihydroxybenzoic acid reverted neurochemical changes and the disrupted oxidative parameters induced by prolonged ethanol exposure (Table 6) [143]. Treatment with this LMWPM was able to revert the diminished choline acetyltransferase (ChAT) activity and the damage to thiobarbituric acid—reactive species (TBA-RS) levels, 2’,7’—dichlorofluorescein (DCFH) oxidation, and superoxide dismutase (SOD) activity provoked by ethanol (Table 6) [143]. Interestingly, the 3,4—dihydroxybenzoic acid showed a protective effect against 6—OHDA–induced dopaminergic neuron loss, a PD model, in zebrafish larvae, when treatment is combined with the flavone chrysin but not isolated (Table 6) [144,145]. These data suggest a synergistic effect between both compounds and, although chrysin is a dietary compound not relevant in circulation like 3,4—dihydroxybenzoic acid, it shows that could be important to explore the effects of LMWPM in conjugation with each other as they circulate in physiological relevant mixtures and not isolated.

In a different approach, 4—hydroxy–3—methoxycinnamic acid derivatives were designed and synthesized as a strategy for the treatment of Alzheimer’s disease (AD). Interestingly, a novel 4—hydroxy–3—methoxycinnamic acid derivative showed a potent neuroprotective effect on Aβ_1-40_—induced larval vascular injury, also promoting favorable dyskinesia recovery rate and response efficiency in AlCl_3_—induced zebrafish AD model (Table 6) [146]. Although these derivatives are not physiologically relevant LMWPM, this study highlighted the potential of zebrafish for revealing their effects in crucial hallmarks of NDDs. Conversely, treatment with high concentrations of sodium benzoate—converted in solution to benzoic acid, induced irreversible neurotoxicity in zebrafish larvae. Sodium benzoate immersion at approximately 6.94 mM significantly reduced tactile sensitivity frequencies of touch—induced larval movement, as well as promoted misalignment of muscle fibers and excess of motor neuron innervations and of acetylcholine receptor (AchR) clusters (Table 4) [147]. Sodium benzoate at 0.28 or 0.7 mM also downregulated expression of tyrosine hydroxylase (TH) and dopamine transporter (DAT) in dopaminergic neurons and impaired larval locomotor activity when at 0.14 or 0.28 mM (Table 5) [148]. Finally, 0.35 mM sodium benzoate induced anxiety—like larval behaviour (thigmotaxis) and oxidative stress by upregulation of glutathione reductase (GSR) expression (Table 5) [149]. These results suggest that some LMWPM when used over a threshold may have pro—oxidant effects that must be taken into consideration. Yet, these concentrations are far higher than those obtained from dietary sources and found in circulation.

Accumulating evidence highlights the neuromodulator functions of dietary parent (poly)phenols, which can provide hints about the putative mode of action of their LMWPM in a nutritional context [121]. The parent (poly)phenol quercetin was protective against the 6—OHDA–induced dopaminergic neuron loss and down—regulated pro—inflammatory gene over—expression (IL—1β, TNF—α, COX—2) in zebrafish larvae [150]. Another example of (poly)phenol that has shown promising results for the treatment of several neurodegenerative diseases, however with poor efficacy and bioavailability, is resveratrol [151]. In zebrafish larvae, resveratrol and alkylated resveratrol prodrugs designed for extended bioavailability were found to recover acetylcholinesterase (AChE) activity from damage by pentylentetrazole, a pro—convulsant competitive GABA antagonist [152]. In adult zebrafish, ellagic acid and curcumin treatment offered protection against rotenone—induced bradykinesia (decreasing locomotor activity) confirming the importance of this model to explore possible therapeutic approaches on movement disorders, including PD [153]. Also (poly)phenol—rich extracts from herbs, fruits and vegetables can be very informative on the mechanism of action of their constituents, although such results need to be interpreted with caution. *Eucommia ulmoides* Oliver, a traditional herb used to treat various diseases, is composed of 28 identified compounds including 3 phenolic acids, 7 flavonoids, and 9 iridoids [154]. Extracts of its leaves significantly reversed the loss of dopaminergic neurons and neural vasculature, as well as lowered the number of apoptotic cells in the zebrafish brain. Notably, locomotor impairments in 1—methyl–4—phenyl–1–1,2,3,6—tetrahydropyridine (MPTP)—modeled PD zebrafish were relieved by this extract in a dose—dependent manner [154].

In conclusion and due to their unique features, zebrafish emerges as an outstanding model for the initial stages of compound—discovery programs targeting the neurological disease. The neurological modulation by the LMWPM in zebrafish is almost unexplored; and opens a new field of research with huge potential to identify and test in vivo relevant molecules, described to circulate in humans, with putative nutraceutical and pharmacological applications on NDDs.

**Table 5 nutrients-13-02940-t005:** Effects of low molecular weight (poly)phenol metabolites in zebrafish neurophysiological functions.

(Poly)phenol Metabolite ^1^	Strain/Start of Exposure	Model without Intervention	Dose and Time	Mechanistic Evidence	Ref.
Benzoic Acids					
Benzoic Acid	AB/2 dpf	Neuronal Teratogenesis	0.07; 0.7; 3.47; 6.94 mM Sodium Benzoate in the Water Medium for 24 h ^2^	↓ Tactile Sensitivity and ↓ Larvae Mobility (6.94 mM) ↑ Misalignment of Muscle Fibers (6.94 mM) Defects on Motor Axons and Neuromuscular Junctions ↑ Axonal Projections and AchR Clusters (post–synapses; >69.4 µM)	[147]
	AB/2 hpf	DA System/Neuronal Development	0.28; 0.94 mM SB in the Water Medium for up to 3 dpf ^2^	↓ TH and DAT Expression in DA Neurons	[148]
	AB/2 hpf	Locomotor Activity	0.14; 0.28 mM SB in the Water Medium for 3 Days ^2^	↓ Locomotor Activity	[148]
	AB/5 hpf	Anxiety–Like Larval Behavior (Thigmotaxis)	0.35 mM SB in E3 Medium for 72 h ^2^	Thigmotaxis ↓ *GSR* Expression	[149]
3,4,5—Trihydroxybenzoic Acid Gallic Acid)	AB/72 hpf	Neuronal Hyperactivity/Motoneuron Hyperexcitability	0.30 mM in E3 Medium for 30 min ^3^	↓ Glutamate and GABA ↑ *fosab* Expression in Distinct Areas of the Brain (Forebrain, Olfactory Bulbs and Pallial Area) ↑ Locomotor Function: Hyperactivity	[142]
	n.d/ 4–6 Month	Neurochemical Content Changes	29.4, 58.8, 117.60 µM in the Water Medium for 24 and 48 h ^3^	↓ Sulfhydryl Content (117.6 µM) ↓ TBA–RS (117.6 µM) ↓ DCFH Oxidation (117.6 µM) ↓ AChE Activity (117.6 µM) = ChAT activity, Nitrates and Nitrites (117.6 µM) ↑ SOD Activity (29.4–58.8 µM) ↑ Catalase Activity (29.4–117.6 µM) = [GSH] (29.4–117.6 µM)	[143]

dpf—days post-fertilization; hpf—hours post-fertilization; Wild—Type Lines: AB/Tuebingen; E3—Zebrafish embryonic medium; DA—Dopaminergic; n.d—non–defined; SB—sodium benzoate; AchR—acetylcholine receptor; TH—Tyrosine hydroxylase; DAT—dopamine transporter; GSR—Glutathione reductase; TBA–RS—thiobarbituric acid–reactive species; DCFH—2′,7′—dichlorofluorescein; AChE—acetylcholinesterase; ChAT—choline acetyltransferase; GSH—glutathione ^1^ (Poly)phenol metabolites are named accordingly the recommendations recently published [32], however the name cited in the original publications where the effect is described is indicated in brackets. ^2^ Converted from mg·L^−1^ to µM using molecular mass of 144.105 g·mol^−1^. ^3^ Converted from mg·L^−1^ to µM using molecular mass of 170.12 g·mol^−1.^ ↑—*increased* ↓—*decreased*.

**Table 6 nutrients-13-02940-t006:** Effects low molecular weight (poly)phenol metabolites in zebrafish models of NDDs.

(Poly)phenol Metabolite ^1^	Strain/Start of Exposure	Model with Intervention	Dose and Time	Mechanistic Evidence	Ref.
Benzoic Acids					
3,4,5—Trihydroxybenzoic Acid (Gallic Acid)	AB/72 hpf	Cd–Induced Disruption of the Olfactory System	293.9 µM in Cd–Treated Larvae in E3 Medium for 30 min ^2^	=Locomotion	[142]
	n.d/ 4–6 Month	Ethanol–Induced Neurochemical Changes	29.4, 58.8 µM in 0.5% (*v*/*v*) Ethanol Solution for 24 h	↑ ChAT Activity (Restored) ↓ TBA–RS (Restored) ↓ DCFH Oxidation (Restored) ↑ SOD Activity (Restored)	[143]
3,4—Dihydroxybenzoic Acid (Protocatechuic Acid)	AB/1 dpf	PD Model Chemical Induced by 6—OHDA	0, 12, 25, 50, 100 µM in the Water Medium for 48 h	= DA Neuron Loss DA Neurons Toxicity at 100 mM	[154]
	AB/1 dpf	PD Model Chemical Induced by 6—OHDA	PCA (6, 12, 25 µM), Chrysin, or PCA + Chrysin in the Water Medium for 48 h	PCA + Chrysin, but not PCA or Chrysin Individually ↓ DA Neuronal Loss	[154]

dpf—days post–fertilization; hpf—hours post–fertilization; Wild–Type Line: AB; E3—Zebrafish embryonic medium; DA—Dopaminergic; 6—OHDA—6—hydroxydopamine; PD—Parkinson’s disease; n.d—non–defined; TBA–RS—thiobarbituric acid–reactive species; DCFH—2′,7′—dichlorofluorescein; ChAT—choline acetyltransferase; SOD—superoxide dismutase. ^1^ (Poly)phenol metabolites are named accordingly the recommendations recently published [32], however the name cited in the original publications where the effect is described is indicated in brackets. ^2^ Converted from mg·L^−1^ to µM using molecular mass of 170.12 g·mol^−1.^ ↑—*increased* ↓—*decreased*.

## 7. Effects of Low Molecular Weight (Poly)phenol Metabolites in Rodent Models of NDDS

Rodents such as mice and rats have been used for decades in research and are by far the most common mammalian animal models in research. This is due to their relatively low dimensions, space requirement and timespan with high breading capability. Mice are amongst the closest relatives to humans apart from primates, scadentia (treeshrews), dermoptera (colugos) and lagomorphs (hares, rabbits, and pikas) maintaining a high degree of similarity to human genes, proteins and cellular pathways.

Protein aggregation, the main feature of all neurodegenerative diseases, has been extensively studied in rodents. In the APP/PS1 mutant mouse model of Alzheimer’s disease, orally administration of 3,4—dihydroxybenzoic acid reduced APP expression and Aβ deposition in hippocampal tissue while increasing learning and memory [155]. Treatment with 3,4,5—trihydroxybenzoic acid also prove to be effective in reducing APP deposition, as it was capable of decreasing the size of amyloid plaques within the hippocampus of APP/PS1 mice [156]. Finally, 3,4,5—trihydroxybenzoic acid administration improved spatial memory of 4–month–old mice while reducing severe deficits on spatial learning and working and reference memory, together with short term recognition of 9–month–old APP/PS1 mice [156].

Data from rodents also showed the anti-inflammatory action of different metabolites. Anti—inflammatory effects of 4—hydroxy–3—methoxycinnamic acid in the lipopolysaccharide (LPS)—induced neuroinflammation and a rotenone—induced rat model of PD have been suggested. In the mentioned models, 4—hydroxy–3—methoxycinnamic acid exerts its effects by inhibiting p—JNK, p—NF—κB, and downstream signaling molecules, such as iNOS, COX—2, TNF—α, and IL—1β, which was accompanied by a remarkable reduction of ionized calcium binding adaptor molecule 1 (Iba—1) and glial fibrillary acidic protein (GFAP) [157,158]. Moreover, 4—hydroxy–3—methoxycinnamic acid treatment interfered with the Toll—like receptor 4 (TLR4)/myeloid differentiation factor 2 (MD—2) complex binding site, crucial for microglial activation and inhibited NF—κB phosphorylation in mouse hippocampus and microglia BV2 cells [157].

Orally administered 4—hydroxy–3—methoxybenzoic acid, as a pretreatment, for two weeks, significantly restored spatial memory. Meanwhile the levels of IL—6, IL—1β, TNF—α, NF—kB p65 subunit and TUNEL positive cells were decreased in the rat hippocampus upon 4—hydroxy–3—methoxybenzoic acid supplementation in the model of transient bilateral common carotid artery occlusion and reperfusion [159,160].

On a model of aged AβPP/PS1 double transgenic mice 3,4—dihydroxybenzoic acid, reduced several inflammatory cytokines such as TNF—α, IL—1β, IL—6, and IL—8, while increasing production of pro—survival signaling factors such as brain—derived neurotrophic factor (BDNF) on the hippocampus and cerebral cortex [155]. In a mouse model of aging using D—galactose, 3,4—dihydroxybenzoic acid decreased NF—kB mRNA expression and protein levels, decreasing COX—2 activity and IL—1β, IL—6, TNF—α and prostaglandin E2 expression in the brain [161]. This compound was also able to significantly reduced microgliosis and astrogliosis in a model of ischemia—induced hippocampal neuronal death [162]. The same was shown by 4—hydroxy–3—methoxybenzoic acid in mice under Aβ intracerebroventricular injection [163]. Aβ injection caused significant impairments in learning and memory tasks in mice, in addition to enhancing pro—inflammatory markers including iNOS and COX—2 expression and production of IL—1β. NF—kB acetylation and nuclear translocation were also examined and found to be significantly enhanced in this model. Pre—treatment with 3,4,5—trihydroxybenzoic acid, however, potently inhibited these effects, likely through suppression of acetylation of NF—kB p65 subunit and NF—kB activity [164].

(Poly)phenols and their metabolites have been reported to combat the effect of oxidative stress in several cell and animal models of neuronal oxidative stress. In a rat model of cadmium—induced heavy metal toxicity, which causes significant disruptions in brain health and cognition, 3,4—dihydroxybenzoic acid oral administration was found to significantly elevate expression of antioxidant enzymes, and reduce markers of oxidative stress and lipid peroxidation, as well as decreasing activity of cholinesterase to preserve cholinergic signaling in the brain [165]. Additionally, intraperitoneally (i.p.) administered 3,4—dihydroxybenzoic acid significantly improved the cognition of aged rats, reduced the content of lipid peroxide, increased the activity of GSH—Px and SOD [166]. 3,4,5—trihydroxybenzoic acid has been proven to have similar neuroprotective effects in the 6—OHDA rat model of PD [167]. The metabolites 3,4,5—trihydroxybenzoic acid and 4—hydroxy–3—methoxybenzoic acid have also been examined in a model of streptozotocin-induced dementia. In this context, these phenolic acids were shown to significantly improve antioxidant defenses and decrease markers of oxidative stress in the brain in addition to attenuating the release of pro-inflammatory cytokines [168,169]. It has been shown that i.p. administration of 4—hydroxy–3—methoxybenzoic acid after Aβ_1-42_—injection enhanced GSH levels and abrogated ROS generation accompanied by an induction of the endogenous Nrf2 and heme oxygenase 1 (HO—1) via the activation of Akt and glycogen synthase kinase 3β (GSK—3β) in the mouse brain [163]. Ren and colleagues investigated protective properties of 4—hydroxy–3—methoxycinnamic acid in hypoxia/ischemia—induced cell injury in vivo and in vitro and revealed that treatment with 4—hydroxy–3—methoxycinnamic acid significantly attenuated memory impairment and reduced hippocampal neuronal apoptosis and oxidative stress in a dose-dependent manner [170]. Treatment with 4—hydroxy–3—methoxycinnamic acid also restored antioxidant enzymes: SOD and CAT, prevented depletion of GSH, and inhibited lipid peroxidation [158]. Moreover, orally administration strongly inhibited mitochondrial apoptotic signaling molecules, such as Bax, cytochrome C, caspase—3, and Poly [ADP-ribose] polymerase 1 (PARP—1), and reversed deregulated synaptic proteins, including postsynaptic density protein 95 (PSD—95), synaptophysin, synaptosomal-associated protein, 25kDa (SNAP—25) synaptosomal—associated protein, 23kDa (SNAP—23), and synaptic dysfunction in LPS—treated mice [170,171]. 4—hydroxy–3—methoxycinnamic acid has also been shown to inactivate the TLR/myeloid differentiation factor 88 (MyD88) pathway [170]. Likewise, 4—hydroxy–3,5—dimethoxybenzoic acid was shown to improve outcomes in the MPTP mouse model of PD, by preserving dopaminergic neurons through activation of antioxidant enzymes and reductions in pro—inflammatory markers [172].

In a mouse model of MPTP induced PD cinnamic acid reduced TH cell death, while increasing dopamine levels and locomotion. Interestingly the authors identified peroxisome proliferator-activated receptor α (PPARα) but not PPARβ as crucial for cinnamic acid effects [173]. Moreover, using a transgenic model of Alzheimer’s disease with cinnamic acid showed reduced plaque formation and increased memory and learning while the same model using knock—out PPARα showed no effects [174]. Also, a mouse PD model with MPTP, 3,4—dihydroxycinnamic acid showed similar results with reduced TH cell death and increased dopamine levels and a reduction of inflammatory markers such as COX—2, TNF—α, and IL—1β together with iNOS and NF—κB [175].

Rats subjected to artery occlusion displayed improvements in markers of injury, showing enhanced neuronal survival, increased SOD and CAT activity, and decreased expression of caspase—3 and 9 when treated with 4—hydroxy–3,5—dimethoxybenzoic acid [176]. 3,4—dihydroxybenzoic acid was shown to significantly improve cognition in a model of global ischemic injury and was found to enhance neuronal survival in the hippocampus of treated animals [162]. These protective effects were associated with increased levels of GSH, and decreased oxidative damage [162]. Several reports have shown similar positive effects on ischemic injury following treatment with 4—hydroxy–3—methoxybenzoic acid (Table 7). These studies indicate that 4—hydroxy–3—methoxybenzoic acid treatment also improves cognition and memory in addition to enhancing neuronal viability, expression of antioxidant enzymes, and long—term potentiation [159,160,177]. The study by Chang and colleagues on AD pathogenesis and associated mechanisms in high—fat diet—induced hyperinsulinemic rats showed that 3,4—dihydroxycinnamic acid enhances SOD glutathione free radical scavenger activity and elevates protein expressions of phosphorylated glycogen synthase kinase 3β (GSK3β), whereas the expression of phosphorylated—tau protein was decreased in the hippocampus of rats administered with 3,4—dihydroxycinnamic acid [178]. Moreover, the expression of APP and β—site APP cleaving enzyme (BACE) were attenuated, subsequently lowering the level of β—amyloid 1–42 (Aβ_1–42_) in the hippocampus of 3,4—dihydroxycinnamic acid treated hyperinsulinemic rats. Moreover, 3,4—dihydroxycinnamic acid also significantly increased the expression of synaptic proteins in high—fat diet rats [178]. Furthermore, this compound was also shown to rescue learning deficits caused by intracranial injection of Aβ_1–42,_ decreasing AChE activity and ROS production together with TNF—α, and IL—6 which was followed by NF—κB and MAPK reduced protein expression [174].

Altogether studies in mice and rats largely contribute to the state of the art regarding the effects of LMWPM inside a mammalian system that better suits the human environment. Due to the presence of a similar gastrointestinal system to humans, mice and rats, these have great potential for studying (poly)phenol metabolism and the role of microbiota. Nevertheless, the use of these animals for cellular mechanistic studies is technically harder than previously mentioned animal models, taking longer periods to achieve and being more expensive. For these reasons, studies addressing LMWPM are still in low numbers, with many injecting doses of (poly)phenols in a concentration far higher than obtainable from dietary sources, in what could be seen has a pharmacological approach.

Overall, only 7 LMWPM were described for the capability to address the main hallmarks of NDDs in different disease models in both mice and rats showing the versatility and potential effects of these (poly)phenol metabolites.

## 8. Conclusions

Over the years, researchers have leveraged a host of different in vivo models in order to dissect neurodegenerative diseases that are commonly heterogeneous in their clinical presentations and are multigenic and multifactorial. In this review we have shown the current evidence of LMWPM on some of the most used gold standard models in science for the study of neurodegenerative diseases: *S. cerevisiae*, *C. elegans*, *Drosophila*, zebrafish, mice and rats. Despite their obvious differences and peculiarities, only the concurrent and comparative analysis of these various model systems will allow the untangling of the effects and mechanisms of action of the LMWPM beneficial effects for the brain dysfunctions described to occur in these diseases. We highlighted the studies that resort to several disease models to study the main hallmarks of neurodegenerative diseases shared amongst Alzheimer’s disease, Parkinson’s disease, Huntington’s disease and others. Yet, only a few LMWPM have been evaluated in these systems. The majority of studies performed to date, have been conducted with dietary flavonoids and other (poly)phenols, not considering their relevant circulating metabolites. Nonetheless, those using LMWPM have shown their potential in different disease models showing their capabilities and versatility, although one should be critical about the doses used to distinguish between their potential dietary relevance or pharmacological approaches.

*S. cerevisiae*, *C. elegans*, *Drosophila* and zebrafish all represent powerful in vivo models in which to perform initial phases of drug screening to identify new candidate compounds, allowing the investigation of potential side effects, through the evaluation of lifespan, embryonic development, adult morphology and behavioral phenotypes, while also giving powerful mechanistic information about the molecular targets of each compound. Meanwhile, mice and rats recapitulate to a higher extend the metabolism of (poly)phenols and LMWPM in humans and the neurodegeneration process occurring in the mammalian brain.

Overall, we expect to observe an increase in the study of LMWPM, both in the screening of unevaluated LMWPM, either alone or in physiological relevant mixtures as they circulate, as well as mechanistic studies on already proven to be beneficial LMWPM. We expect to reach a more comprehensive and powerful understanding of the effects of LMWPM from diets rich in (poly)phenols, like the Mediterranean diet, and potentially finding powerful molecules to prevent and help the treatment of neurodegeneration and neurodegenerative diseases.

## Figures and Tables

**Figure 1 nutrients-13-02940-f001:**
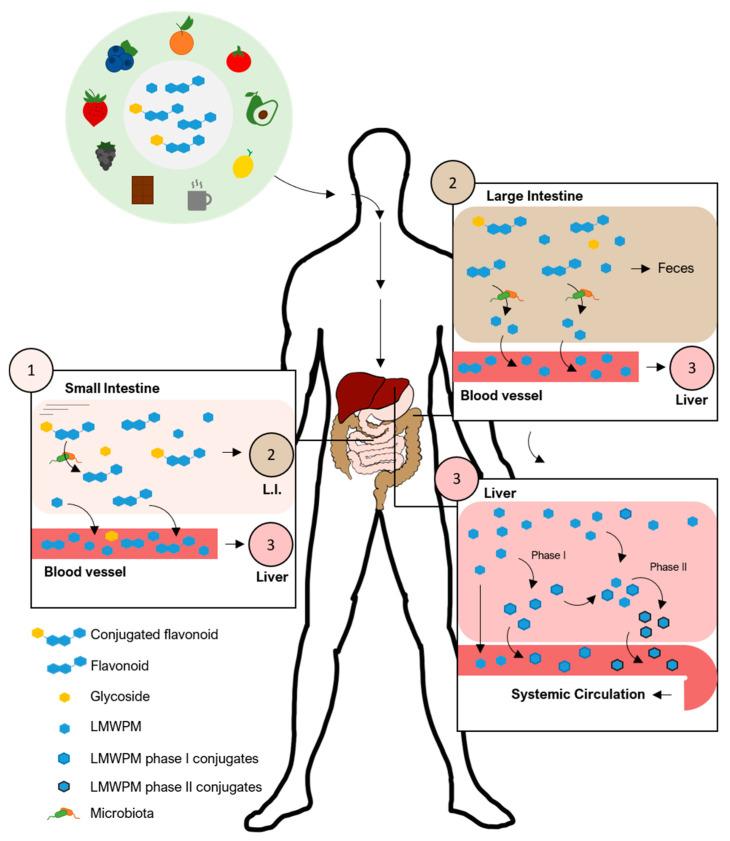
Diets rich in (poly)phenols originate large amounts of flavonoids and flavonoid conjugates reaching the digestive system. (1) In the small intestine, epithelial and bacterial enzymes remove sugar conjugates while absorption occurs. (2) The majority of flavonoids travel to the lower part of the gut where microbiota catabolizes flavonoids into LMWPM that will be absorbed into enterohepatic circulation. (3) LMWPM will enter systemic circulation intact or will undergo phase I and II metabolic reactions by intestinal and liver cells before reaching the circulation. LMWPM—low molecular weight (poly)phenol metabolites. L.I.—large intestine.

**Figure 2 nutrients-13-02940-f002:**
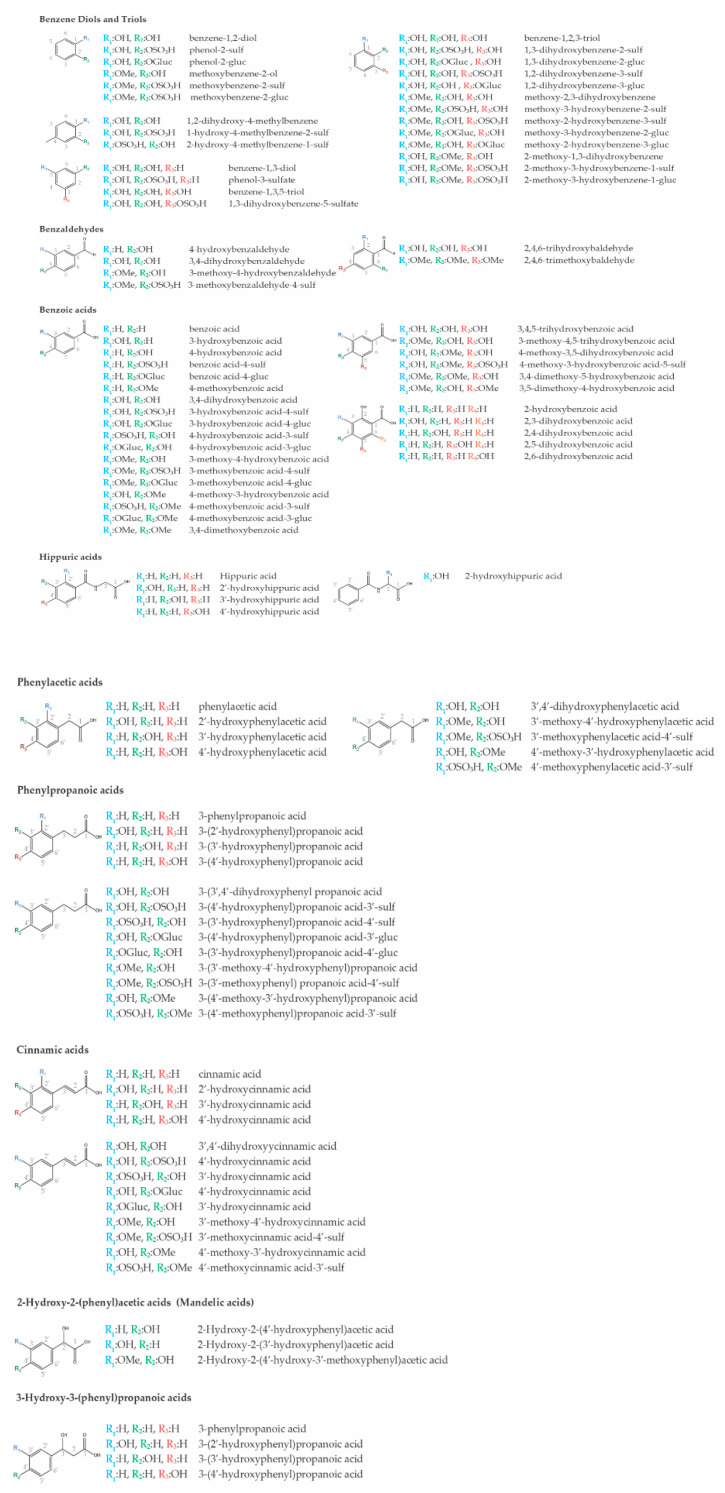
Chemical structures of low molecular weight (poly)phenol metabolites resulting from gut microbiota catabolism of flavonoids or enterohepatic phase I and II metabolic conversions. The nomenclature used followed recent recommendations [32]. Sulf—sulfate; Gluc—glucuronide; Me—methyl group.

**Figure 3 nutrients-13-02940-f003:**
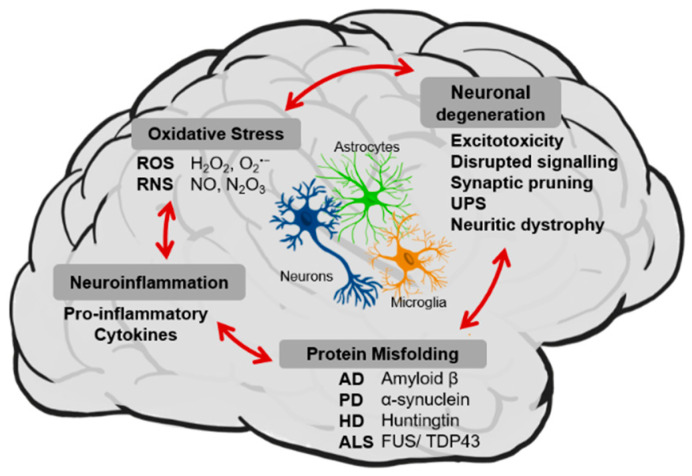
Main hallmarks of neurodegeneration and neurodegenerative diseases—protein misfolding, neuroinflammation, neuronal degeneration and oxidative stress—shared between the major cellular types present in the brain such as neurons, astrocytes and microglia. UPS: ubiquitin—proteosome system, ROS: reactive oxygen species, RNS: reactive nitrogen system, AD: Alzheimer’s disease, PD: Parkinson’s disease, HD: Huntington’s disease, ALS: amyotrophic lateral sclerosis.

**Table 4 nutrients-13-02940-t004:** Effects of low molecular weight (poly)phenol metabolites on models of neurodegeneration in *Drosophila.*

(Poly)phenol Metabolite ^1^	Strain	Model	Dose and Time	Mechanistic Evidences	Ref.
Benzoic Acid					
3,4,5—Trihydroxybenzoic Acid (Gallic Acid)	WT Canton–S	PD Chemical Model Induced by PQ	0.1 mM in 0.1% Sucrose	↑ Lifespan ↑ Locomotor Activity	[106]
	Tg (TH—GAL4: UAS—Dmp53 RNAi)Canton–S Background	PD Chemical Model Induced by PQ	0.1 mM	↑ Life Span ↑ Locomotor Activity	[107]
	Parkin Knockout Tg (TH—Gal4: UAS—ParkinRNAi)	PD Genetic Model Chronically Exposed to PQ	0.1 or 0.5 mM Alone or With PQ	No Rescue in Lifespan or Locomotor Activity	[108]
	Tg (UAS:hAPP, UAS:BACE1)	AD Genetic Model	50 or 100 µM	↑ Catalase ↓Malonaldehyde ↓AChE	[109]
	WT Oregon–K	Urethane Exposure Genotoxic Environmental Carcinogen Model	5.88 M + Urethaen 20 mM in 10% Sucrose for 72 h ^2^	↑ Antigenotoxic and Antioxidant Role ↑ Glutathione S–Transferase ↑ SOD ↑ Catalase	[105]
Cinnamic Acid					
4—Hydroxy–3—Methoxycinnamic Acid (Ferulic Acid)	wt Canton–S	pd Chemical Mode l20 mm pq Exposure for 24 h	0.1 mM in 0.1% Sucrose	↑ Lifespan ↑ Locomotor Activity	[106]
3,4—Dihydroxycinnamic Acid (Caffeic Acid)	wt Canton–S	pd Chemical Model of pq Exposure for 24 h or 48 h	0.5 and 1 mM in 0.1% Sucrose	↑ Lifespan ↑ Locomotive Activity	[106]
	WT W^1118^		1.1 mM to 5.55 mM ^3^	↑ Lifespan ↑ Climbing Behavior ↑ Stress Resistance (Heat and Starvation). ↑ Mitochondrial Function ↑ Antioxidant Capacity	[110]

WT—wild type; PD—Parkinson’s disease; PQ—paraquat; SOD—Superoxide dismutase. ^1^ (Poly)phenol metabolites are named accordingly the recommendations recently published [32] however the name cited in the original publications where the effect is described is indicated in brackets. ^2^ Converted from w/v to mM using molecular mass of 170.12 g·mol^−1^. ^3^ Converted from mg·L^−1^ to mM using molecular mass of 180.16 g·mol^−1.^ ↑—*increased* ↓—*decreased*.

**Table 7 nutrients-13-02940-t007:** Effects low molecular weight (poly)phenol metabolites in mice and rat models of NDDs.

(Poly)phenol Metabolite ^1^	Species	Model	Dose and Time	Mechanistic Evidence	Ref.
Benzoic acids					
4—hydroxy–3—methoxybenzoic Acid (Vanillic Acid)	Sprague-Dawley rat	Cerebral Ischemia-Reperfusion (I/R)	50 and 100 mg/kg Gavage for 14 Days	↓ NF—κB p65 Protein Levels ↓ IL—6, IL—1β and TNF—α	[179]
	C57BL/6N Mice	Intracerebroventricular (i.c.v.) Injection of Aβ_1-42_	30 mg/kg i.p. for 3 Weeks	↓ Number of Activated Microglia and Astrocytes Cells ↓ iNOS Expression ↓ NF—kB	[163]
	Wistar Rat	Carotid Artery Occlusion and Reperfusion in Rat	100 mg/kg, Gavage for 14 Days	↓ IL—6, TNF—α and TUNEL Positive Cells. ↑ IL—10 Levels in the Hippocampus	[177]
	Swiss Albino Mice	Streptozotocin (STZ)–Induced Neurodegeneration	25, 50, and 100 mg/kg Orally for 28 Days	↑ Spatial Learning and Memory Retention ↓ Oxidative Stress ↓ AChE, Corticosterone, TNF—α ↑ SOD, CAT and GPx Levels	[169]
	Wistar Rat	Transient Bilateral Common Carotid Artery Occlusion and Reperfusion	10, 30, 100 mg/kg Gavage for 2 Weeks	Attenuation of Reactive Hyperemia and BBB Disruption. ↑ Sensory Motor Signs and Anxiolytic Behavior.	[177]
	Wistar Rat	Transient Bilateral Common Carotid Artery Occlusion and Reperfusion	100 mg/kg Gavage for 2 Weeks	↑ Locomotion and Memory ↓ Cell Death of CA1 Neurons	[159]
	Wistar Rat	Transient Bilateral Common Carotid Artery Occlusion and Reperfusion	100 mg/kg Gavage for 2 Weeks	Restoration of the Spatial Memory ↓ IL—6, TNF—α and TUNEL Positive Cells ↑ IL—10 Levels in the Hippocampus	[160]
	Sprague-Dawley	Middle Cerebral Artery Occlusion	50 and 100 mg/kg Gavage for 14 Days	↓ Levels of Lipid Peroxidation ↓ Malondialdehyde ↑ SOD and CAT	[179]
3,4—Dihydroxybenzoic Acid (Protocatechuic Acid)	AβPP/PS1	Animal Transgenic Model	100 mg/kg Orally for 4 Weeks	↓ Aβ Deposition ↓ Tnf—A, Il—1β, Il—6 And Il—8 Expression. ↑ Bdnf ↑ Learning and Memory Performance	[155]
	Balb/cA mice	D—Galactose	0.5%, 1% or 2% Enriched Food Content for 8 Weeks	↓ IL—1β, TNF—α, IL—6 and Prostaglandin E2 ↓ COX—2 Activity and Expression ↓ mRNA Expression and Protein Production of NF—kB p65	[161]
	Sprague-Dawley rat	Ischemia–Induced Hippocampal Neuronal Death	30 mg/kg/day Orally for 7 Days	↓ Microglial Activation ↓ Astroglial Activation	[162]
	Wistar rat	Cadmium–Induced Neurotoxicity	10 and 20 mg/kg Orally for 21 Days	↓ Na^+^/K^+^–ATPase Activity ↓ Acetylcholinesterase, Butyrylcholinesterase and Endogenous Antioxidant Enzymes	[165]
	Sprague-Dawley rat	Ischemia–Induced Hippocampal Neuronal Death	30 mg/kg/day Orally for 7 Days	↓ Neuronal Cell Death ↓ Oxidative Stress ↓ BBB Disruption	[162]
	AβPP/PS1 mice	Animal Transgenic Model	100 mg/kg orally for 4 Weeks	↓ Aβ Deposition in Hippocampal Tissue ↓ APP Expression	[155]
3,4,5—Trihydroxybenzoic Acid (Gallic Acid)	FVB Mice	Kainate-Induced Neuronal Injury	1 mg/kg by Gavage for 3 Days	↓ Lipid Peroxidation In Vivo ↓ Ca^2+^ Release, ROS, Lipid Peroxidation, Expression of COX—2 and p38 MAPK	[180]
	Wistar Rat	Oxidative Stress Induced by 6—OHDA	50, 100 and 200 mg/kg Oral Gavage for 10 Days	↑ Levels of GSH, SOD and CAT ↓ NO Concentration. ↓ TNF—α, IL—1β and IL—6.	[167]
	Wistar Albino Rat	Intracerebroventricular Injection of Streptozotocin (STZ)	30 mg/kg Oral Gavage for 26 Days	↑ Total Thiol Content ↑ GSH–Px, SOD and CAT Enzyme Activities	[168]
	APP/PS1 Mouse Model of AD	Animal transgenic model	30 mg/kg/day administration through gavage for 30 days	↓ Aβ_1–42_ Plaque Size in Hippocampus and Cortex	[156]
4—Hydroxy–3,5—Dimethoxybenzoic Acid (Syringic Acid)	C57BL/6 Mice	MPTP and probencid (MPTP/p)	20 mg/kg orally or oral gavage for 35 days	↑ Motor Functions ↑ TH, DAT and VMAT2 in SN	[172]
	C57BL/6 Mice	Artery occlusion ischemic injury	10 mg/kg i.p. injection at the time of occlusion	↑ SOD ↑ Nuclear Respiratory Factor 1 Levels ↓ Caspase—3 and—9 Levels	[176]
Cinnamic acids					
Cinnamic acid	C57BL/6 Mice WT and PPARα (-/-)	MPTP Induced Parkinson’s Disease	100 mg/ kg Orally For 7 Days	↓ Neuronal Cell Death ↑ Dopamine ↑ Locomotion ↑ PPARα	[173]
	B6SJL-Tg (5xFAD)	Trangenic Model of Ahlzeimer‘s Disease	100 mg/kg	↑ Locomotion and Memory ↓ Plaque Formation and Aβ_1–40_ ↑ Lysosomal Biogenesis	[174]
4—Hydroxy–3—Methoxycinnamic Acid (Ferulic Acid)	C57BL/6N mice	LPS	20 mg/kg Orally for 11 Days	↓ Glial Cell Activation ↓ p—JNK, p—NF—κB ↓ iNOS, COX—2, TNF—α, and IL—1β in the Mouse Hippocampus and BV2 Microglial Cells	[157]
	Sprague-Dawley Rat	Hypoxia/Ischemia-Induced Cell Injury	28, 56 and 112 mg/kg for 5 Days	↓ Apoptosis ↓ Cleaved Caspase—3 and Bax ↑ Bcl—2 ↓ ROS Generation ↓ Ca^2+^ Influx ↓ Superoxide Anion (O^2−^) ↓ Malondialdehyde ↑ SOD and GSH-Px Activity	[170]
	Wistar Rat	Rotenone Induced Parkinson’s Disease	50 mg/kg i.p. for 4 Weeks	↓ Cell Death ↓ Lipid Peroxidation ↓ Malondialdehyde Content ↑ SOD, CAT and GSH	[158]
3,4—dihydroxycinnamic acid (Caffeic Acid)	Sprague- Dawley Rat	High—Fat Diet—Induced Hyperinsulinemia	30 mg/kg/Daily Oraly for 30 Weeks	↑ Sod And Gsk3β ↓ P—Tau ↓ App, Bace And Aβ _1−42_ ↑ Levels Synaptic Proteins ↓ Memory Deficits	[178]
	Swiss Albino Mice	Rotenone Induced Parkinson’s Disease	2.5, 5 or 10 mg/kg	↑ Dopamine Concentration ↓ COX—2, TNF—α, and IL—1β, CD11b ↓ iNOS, NF—κB ↓ Neuronal Cell Death	[175]
	Sprague-Dawley Rat	Intracranial Injection of Aβ_1-40_	100 mg/kg Injection for 2 Weeks	↑ Learning ↓ AChe, ROS ↑ CAT, GSH ↓ TNF—α, IL—6 ↓ NF—κB, Caspase—3, p—53	[181]

Aβ—Amyloid beta; CAT—catalase; NF–κB—Nuclear factor kappa–light–chain–enhancer of activated B cell; GSK—Glycogen synthase kinase; GSH—glutathione; APP—Amyloid precursor protein; ROS—Reactive Oxygen Species; TNF—Tumor necrosis factor; JNK—c–Jun N–terminal kinases; *SOD*—Superoxide Dismutase; *Tg*—transgenic; PPAR—Peroxisome proliferator–activated receptor alpha; LPS—lipopolysaccharide; MPTP—1—methyl–4—phenyl–1,2,3,6—tetrahydropyridine. ^1^ (Poly)phenol metabolites are named accordingly the recommendations recently published [32], however the name cited in the original publications where the effect is described is indicated in brackets. ↑—*increased* ↓—*decreased*.

## Data Availability

Not applicable.

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
