# Peer review of "Overview of Beneficial Effects of (Poly)phenol Metabolites in the Context of Neurodegenerative Diseases on Model Organisms"

_nutrients, 2021, doi:10.3390/nu13092940_

Round 1
Reviewer 1 Report
In this review, the authors discuss different model organisms that have been used to study the beneficial effects of low molecular weight (poly)phenol metabolites (LMWPM) for neurodegenerative diseases (NDDs). The review is important because it can help the study of LMWPM on NDDs, both in the screening of unevaluated LMWPM, either alone or in physiological relevant mixtures as they circulate, as well as mechanistic studies on already proven to be beneficial LMWPM, as the authors described.
Here are my comments:
- Regarding the bioavailability of polyphenols, it should be mentioned that polyphenols have a high absorption rate. Taking into consideration the parent compound and the metabolites from colon bacteria catabolism, more than 80% of a dose can be absorbed and ultimately excreted in the urine. (DOI: 10.1111/nbu.12278; DOI: 10.1124/dmd.109.030304; DOI: 10.1016/j.freeradbiomed.2012.05.023).
- I suggest that the order of Table 1 and Table 2 should be reversed with the related descriptions. First, describe the studies related to LMWPM in S. cerevisiae models of NDDs (Table 2). Then describe some related studies on oxidative damage of LMWPM (Table 1), because the latter is not suitable for direct classification as models of NDDs.
- S. cerevisiae; C. elegans in lines 27, and S. cerevisiae in line 770 should be changed to italics.
- The spacing of many sentences is not uniform
Author Response
Thank you very much for the time dedicated to the review of this article and the helpful and constructive criticism to produce a higher quality manuscript.
Regarding your comments:
Point 1: Regarding the bioavailability of polyphenols, it should be mentioned that polyphenols have a high absorption rate. Taking into consideration the parent compound and the metabolites from colon bacteria catabolism, more than 80% of a dose can be absorbed and ultimately excreted in the urine. (DOI: 10.1111/nbu.12278; DOI: 10.1124/dmd.109.030304; DOI: 10.1016/j.freeradbiomed.2012.05.023).
Answer: This information was incorporated in the manuscript in section 2.
Point 2: I suggest that the order of Table 1 and Table 2 should be reversed with the related descriptions. First, describe the studies related to LMWPM in S. cerevisiae models of NDDs (Table 2). Then describe some related studies on oxidative damage of LMWPM (Table 1), because the latter is not suitable for direct classification as models of NDDs.
Answer: The order of Table 1 and Table 2 was changed, together with the text, as suggested.
Point 3: S. cerevisiae; C. elegans in lines 27, and S. cerevisiae in line 770 should be changed to italics.
Answer: This concern has been revised and correct across the manuscript.
Point 4: The spacing of many sentences is not uniform.
Answer: This concern has been revised and correct across the manuscript.
Thank you very much,
Diogo Miguel José Carregosa
PhD Student, CEDOC-Nova Medical School
Reviewer 2 Report
The review by Carregosa et al represents an interesting review of the activity of low molecular weight phenols in various experimental models. Mostly, the review is rich, well organized and reflects a significant effort in navigating and reviewing the literature. I find the manuscript worth publishing after attending to some considerable changes:
-The section on yeast model is too long and full of details contrary to the section on clinical studies which is too short and only highlights the drawbacks of clinical trials. The authors may consider changing the title and moving the section on clinical trials to the introduction and reconsider their main focus on non-human models.
-The title stresses that the models will go from single cell to the whole human body but the manuscript is essentially focusing on non-human models
-There is no mentioning of cell line research as if it does not exist. This is a major pitfall.
However, I find that in few instances the use of English language needs to be reviewed to avoid redundancy, simplify sentences and correct punctuation.
Examples of sentences needs to be re-written or corrected:
Please review all binomial names and make sure they are all written in italics.
Review punctuation carefully
Line 22-23: Notwithstanding …
Line 34: thus rich in …
Line 54: becoming less …
Line 62: delete “reaching the gut” informal and redundant after using GIT
Line 65-67: consider rewriting to simplify and improve the flow
Line 81-82: fix “longer periods time “
Line 100: correct Lewis to Lewy
Line 101: amyotrophic should be lowercase
Line 411: insert “acids” after “cinnamic”
Line 432: NDDsmodels use spaces
Line 474-475: rewrite the meaning is vague
Line 532-533: “lifts the veil …” consider rewriting
Line 626: in what?
Line 714: Ache change to AChE
Other points to consider/correct:
Figures 1, 2 and 3: Please improve their resolution, difficult to read
In Figure 2: Chemically speaking, “benzenes” is not the correct term. Once the hydroxyl group introduced to the benzene ring, they are “phenols” that have distinctive chemical, metabolic and biological characteristics. None of the compounds enlisted is a benzene, they are phenols. If there is no free OH group, they are not biologically active and they start showing toxic effects.
Line 244: benzene 1,3,5-triol is the correct IUPAC nomenclature. Commas to be used between numbers
Line 463: convert mg/mL to molar concentration to be consistent with all other concentrations used in the manuscript
Line 572-573: “is composed of 28 compounds …” this needs to. Be corrected botanical extracts contain 200-300 compounds on average. The 28 compounds here are the ones that could be identified. Please rewrite to provide the correct meaning.
Line 683: cinnamic acid itself is not a phenol and it I metabolised in humans to hippuric acid. I find this part should be omitted.
Page 23 of 36: cinnamic acid is not a phenol, consider removing this part
Author Response
Thank you very much for the time dedicated to the review of this article and the helpful and constructive criticism to produce a higher quality manuscript.
Regarding your comments:
Point 1: The section on yeast model is too long and full of details contrary to the section on clinical studies which is too short and only highlights the drawbacks of clinical trials. The authors may consider changing the title and moving the section on clinical trials to the introduction and reconsider their main focus on non-human models.
Answer: The section on clinical trials was moved to the introduction, the focus changed to non-human models and the title changed.
Point 2: The title stresses that the models will go from single cell to the whole human body but the manuscript is essentially focusing on non-human models
Answer: The tittle was changed, as suggested.
Point 3: There is no mentioning of cell line research as if it does not exist. This is a major pitfall.
Answer: This issue was addressed. Other publications have already reviewed the evidence of (poly)phenol metabolites in cell lines. Such publications have been included.
Point 4: However, I find that in few instances the use of English language needs to be reviewed to avoid redundancy, simplify sentences and correct punctuation.
Answer: The language used in the manuscript was revised.
Point 5: Examples of sentences needs to be re-written or corrected.
Answer: Such examples have been corrected.
Point 6: Please review all binomial names and make sure they are all written in italics.
Answer: All binomial names have been revised and corrected.
Point 7: Figures 1, 2 and 3: Please improve their resolution, difficult to read
Answer: All figures have been redone and exported to improve their resolution.
Point 8: In Figure 2: Chemically speaking, “benzenes” is not the correct term. Once the hydroxyl group introduced to the benzene ring, they are “phenols” that have distinctive chemical, metabolic and biological characteristics. None of the compounds enlisted is a benzene, they are phenols. If there is no free OH group, they are not biologically active and they start showing toxic effects.
Answer: The word “benzenes” has been change to “benzene diols and triols” to follow the most recent recommendations adopted by the peers in (poly)phenol research (https://doi.org/10.1093/ajcn/nqaa204).
Point 9: Line 244: benzene 1,3,5-triol is the correct IUPAC nomenclature. Commas to be used between numbers
Answer: Revised and corrected.
Point 10: Line 463: convert mg/mL to molar concentration to be consistent with all other concentrations used in the manuscript
Answer: Revised and corrected.
Point 11: Line 572-573: “is composed of 28 compounds …” this needs to. Be corrected botanical extracts contain 200-300 compounds on average. The 28 compounds here are the ones that could be identified. Please rewrite to provide the correct meaning.
Answer: The phrase was rewritten to clarify this issue.
Point 12: Line 683: cinnamic acid itself is not a phenol and it I metabolised in humans to hippuric acid. I find this part should be omitted.
Answer: Although cinnamic acid is not by IUPAC classification a phenolic, due to the lack of a hydroxyl group in the benzene ring, cinnamic acid has been classically grouped with the remaining “cinnamic acids” in the study of (poly)phenol metabolites due to its presence being related with microbiota catabolism of flavonoids. In fact, there is the possibility that cinnamic acid is generated through microbiota or by phase I conversion of 2-hydroxycinnamic acid, 3-hydroxycinnamic acid and/or 4-hydroxycinnamic acid.
Cinnamic acid can in fact be metabolized to hippuric acid however, during this process it is converted to other LMWPM classes. Cinnamic acid can be converted to phenylpropanoic acids, followed by phenylacetic acids, benzaldehydes, benzoic acids and then benzoic acid is conjugated with glycine forming hippuric acid (https://doi.org/10.1021/acs.jafc.9b02155). In fact this metabolic conversion occours to all LMWPM present in the review along time in circulation.
Due to this reasons, and the most recent recommendations in (poly)phenol metabolite research to continue to classify cinnamic acid as one of “cinnamic acids” (https://doi.org/10.1093/ajcn/nqaa204) we kept this metabolite in the manuscript.
Point 13: Page 23 of 36: cinnamic acid is not a phenol, consider removing this part
Answer: See previous answer to point 12. Cinnamic acid was kept due to recent recommendations in (poly)phenol metabolite research (https://doi.org/10.1093/ajcn/nqaa204)
Thank you very much,
Diogo Miguel José Carregosa
PhD Student, CEDOC-Nova Medical School
Round 2
Reviewer 2 Report
Happy to recommend to the editor for publication
Author Response
Dear Reviewer 2,
Thank you very much for the time dedicated to the revision of this manuscript and the high valuable input to increase its overall quality.
Thank you once more,
Diogo Carregosa, PhD Student, CEDOC, NOVA Medical School